# From anxiety to action—Experience of threat, emotional states, reactance, and action preferences in the early days of COVID-19 self-isolation in Germany and Austria

**Stefan Reiss**[1]*, **Vittoria Franchina**[1○], **Chiara Jutzi**[1○], **Robin Willardt**[2○], **Eva Jonas**[1]

**1** Department of Psychology, University of Salzburg, Salzburg, Austria, **2** Department of Psychology, ETH Zurich, Zurich, Switzerland

○ These authors contributed equally to this work.
* stefan.reiss@sbg.ac.at

**Data Availability Statement:** The data underlying the results and the R analysis script are available in

## Abstract

The COVID-19 pandemic has interrupted everyday life virtually everywhere in the world, enabling real-life research on threat-and-defense processes. In a survey conducted within the first days of implementing social distancing measures in Austria and Germany, we aimed to explore the pathways from threat perception to preferences of defense strategies. We found that anxiety, approach-related affect, and reactance were specifically elicited by motivational (vs. epistemic) discrepancies. In a second step, we tested the mediating effect of anxiety, approach-related affect, and reactance on preferences regarding personal-social and concrete-abstract defenses. Experiencing anxiety was related to interest in security-related actions, and approach-affect was related to both personal projects and social media use. Participants experiencing reactance were more inclined to pursue personal projects (personal-abstract) and less interested in security-related (personal-concrete) actions. They also showed marginally lower system justification (social-abstract). Additionally, we examined the relationship of loneliness with defense strategies, showing that loneliness was associated with lower system justification and security behaviors. The results suggest that individuals deal with threat in their own ways, mostly depending on affective state and motivational orientation: Anxiety was related to security, approach-state to action (both social and personal), reactance to derogation of the system and disregard for security, while loneliness was associated with inaction.

## Introduction

With the worldwide outbreak of the novel coronavirus, national decision-makers were challenged to act quickly to contain the spread of the virus: they had to keep residents safe and healthy without stunting economies. Although the steps taken (e.g., curfews and restrictions) varied across countries, the situation has led to some astonishingly similar reactions. Fearing infections, many people turned to security-related behaviors such as increased hand hygiene and complied with the restrictions put into place. Some started engaging in new projects,

the supporting information files attached to the submission.

**Funding:** Our research was funded by the Austrian Science Fund (FWF-P27457; https://www.fwf.ac.at/en/). CJ and VF were supported by the Doctoral College "Imaging the Mind" of the Austrian Science Fund (FWF-W1233-B). The funders had no role in study design, data collection and analysis, decision to publish, or preparation of the manuscript.

**Competing interests:** The authors have declared that no competing interests exist.

sports, learning new skills, or engaging in social media to stay connected. Others seemed reactant in the face of the restrictions, voiced their concerns about personal freedoms, and showed their willingness to ignore curfews and social distancing measures [1, 2]. This is potentially harmful on a personal and societal level and urges for investigations of the psychological processes driving COVID-19-related reactions. To deal with a global disease outbreak efficiently and adequately, we need to harness (social) psychological knowledge and try to understand human experience and behavior. By doing so, we can ultimately improve the public response to pandemics (for an overview, see [3]).

We will address this issue from a theoretical perspective provided by research on psychological threat and defense [4]. Several psychologists have described the global pandemic as psychologically threatening [5–7] with the potential to arouse anxiety and shape defensive behavior. The present research pursues several goals: Firstly, we aim to investigate whether the COVID-19 pandemic was perceived as a psychological threat through subjective epistemic (meaning-related) discrepancies, or through motivational-affective discrepancies (i.e., lack of control, agency, efficacy) in Austria and Germany. This allows us to identify the nature of the threat imposed by COVID-19. Secondly, we aim to explore the pathways predicting preferences for certain defensive strategies through COVID-19-induced affective states. Not only does this research investigate anxiety- and approach-related affect, but also state reactance as motivated response to subjectively unjust freedom restrictions, and loneliness as a circumstance that may catalyze social defense strategies.

## Psychological threat and defense

From the perspective of threat-and-defense research, the current situation is a combination of concerns regarding psychological needs. This pandemic being a very concrete and situational interruption of everyday life, there are two main categories of discrepancies possibly responsible for threat perceptions: motivational and affective discrepancies related to goal-attainment [8], and epistemic discrepancies related to understanding and predicting situations [9]. According to the General Process Model of Threat and Defense Model (GPM) [3], such discrepancies reduce ongoing goal-oriented approach motivation and elicit a state of anxiety and arousal that is driven by the Behavioral Inhibition System (BIS) [10, 11]. In order to reduce BIS-related symptoms such as anxiety and uncertainty, individuals can engage in approach-related defense strategies mediated by the Behavioral Activation System (BAS) [10]. These *distal defenses* can vary across two dimensions: they can be oriented toward oneself (personal) or the social environment (social); they can also be concrete (i.e. trying to resolve the threat or described as detailed action plans) or abstract (more value-oriented and related to diffuse long-term goals). The combination of these two dimensions of defensive strategies has been proposed by the GPM [4]. However, attempts to predict the choices and preferences for types of strategies have been scarce. We assume that the motivational-affective state elicited, that is, anxiety-related inhibition (BIS) or approach-related activation (BAS) drive preferences and intentions for subsequent behavior. However, due to the situational restrictions in everyday life, a third motivational-affective state may play into the dynamic, namely reactance. Regaining agency and re-establishing personal freedom after threat can be facilitated by reactance [12]. The sudden situational restrictions of individual freedoms to protect at-risk individuals have been met with acceptance by most but with resistance by others. The perceived restriction of personal specific freedoms can to behavior aiming to restore or protect the threatened freedom [13]. We assume that to some individuals, these restrictions elicit feelings of reactance, which in turn increase willingness to circumvent curfews and rule-following. Likewise, feelings of being treated unjustly should manifest in decreased system justification.

## Defensive strategies

In this study, we aim to explain and predict preferences for ways to deal with the coronavirus situation in four different ways: by engaging in personal or social strategies that are either concrete or abstract. All these strategy categories are assumed to help moving from an initial stage of anxious uncertainty to agentic goal-drive behavior [4]. Based on observed reactions to the restrictions COVID-19 has put on everyday life in Germany and Austria, we proposed four categories of defenses referring to the personal-social and concrete-abstract dimensions.

Investment in *personal projects* is a *personal abstract* defense strategy: pursuing personally relevant goals, doing sports, and having a creative outlet is self-enhancing without direct engagement with the threatening situation [14].

*System Justification* is a *social defense*, and an *abstract* way of fortifying worldviews [15, 16]. System justification has been shown to increase after threat experience [17]. News coverage of the COVID situation seems to anecdotally confirm this assumption [18, 19], and increased in nationalism have been observed and theorized [20]. On the other hand, Knight et al. [21] summarized people high in trait reactance were more drawn to fighting the system rather than justifying it.

*Social Media Usage* is a concrete social defense, re-establishing social contacts immediately. The internet has been the prime source of information on current events surrounding the pandemic [22]. Aside from providing quick and easy-to-access information (and, of course, misinformation), people can turn to social media to find a direct social remedy to the individual threat imposed by COVID-19.

*Security efforts* are concrete ways to increase safety and reduce infection risks. In Germany, sales of soap and disinfectant have risen dramatically since the outbreak of the pandemic [23]. Strict hygiene and following social distancing guidelines serve directly to reduce infection risks for oneself and others and increase personal perceptions of security. BIS-anxiety should therefore predict personal security efforts.

Additionally, we aimed to investigate the influence of loneliness on preferences for defensive strategies. Many people have been forced to live in self-isolation or to reduce social contacts, leading to feelings of loneliness [24, 25]. Loneliness is the painful experience of emotional and social disconnect [26, 27], which has shown to increase interest in social interactions [24]. Therefore, we hypothesize that individuals experiencing loneliness should engage more in social interactions through social media [28], and should be less concerned about social distancing, reporting lower levels with regard to safety-related behaviors. Moreover, loneliness is an assumed predictor for behavioral disengagement [29]. Thus, people experiencing loneliness might be less motivated to pursue personal projects [30]. In addition, they might be more prone to fight the system, blaming it as the cause of their current loneliness. Following this line of thoughts, we expect loneliness to be negatively related to system justification, security-related efforts, and personal projects, but positively related to social media use.

## Present research

In this study, we aimed to investigate different hypotheses regarding the social psychological aspects of this interruption of every-day life as we knew it. Our first aim was to assess the nature of discrepancies experienced in the early days of isolation. During the first days of implementing corona-related curfews and stay-at-home policies, the general perception of the situation—in our view—was as a concrete and situational threat. As such, we aimed to gauge threat perception as a latent variable originating from either *motivational-affective discrepancies*, that is, thwarted feelings of control, agency, and aversive frustration of clarity, or from epistemic needs (feeling of surprise or unexpectedness). Thus, the analyses sought to compare

motivational vs. cognitive discrepancies as sources of threat, concretely applied to the early
COVID-19 stages. We also aimed to illuminate the threat-related affective state of individuals
in response to these perceived discrepancies, by assessing inhibited anxious uncertainty (BIS-
related affect), reactance as response to unjust freedom restrictions, and agentic approach-
related affect (BAS-related). We assumed that perceptions of motivational discrepancies would
relate to affective states, which would in turn predict the preference for specific actions in the
corona situation: Discrepancy-induced anxiety should be related to increased attractivity of
social media use as well as security-related behaviors (e.g., washing hands and stockpiling),
while approach-oriented individuals should prefer pursuing personal projects (e.g., being crea-
tive, doing sports) [8]. Individuals with high amounts of reactance should have decreased pref-
erence for security-related behavior and be more inclined to break curfews installed in the
early days of self-isolation [2]. Our second aim was to find states and perceptions related to
support for national decision-makers and system justification. Here, we assumed that system
justification would be specifically related to reactance.

## Materials and procedure

### Participants

Ethical approval for the research was granted by the ethics committee of the Paris Lodron Uni-
versity Salzburg and participants gave their informed consent to participate in the study. The
research was promoted as an online study investigating feelings and perceptions in the context
of the coronavirus pandemic. Participants were recruited through social media and online
news promoting the survey link. In the first days of the isolation (March 20–26, 2020), a total
of 404 participants completed the survey, stopping data collection at that point to not dilute
the momentary assessment. Filtering for participants currently in Germany and Austria, nine
participants were excluded, leaving a final sample of 395 participants (126 from Austria, 269
from Germany). Mean age was 34.4 years (range 14–78, $SD$ = 14.8), with 267 female partici-
pants (128 male). 91 participants had general qualification for university entrance, 125 had
general qualification without a finished university education; 179 had finished at least a univer-
sity degree (e.g., Bachelor's degree). For subsequent analyses, a dichotomous variable was cre-
ated (university degree = 1; no degree = 0). 51 respondents (13%) considered themselves part
of a risk group, and 48 respondents reported existing health precondition (respiratory diseases,
autoimmune diseases, heart conditions or similar). Data were analyzed after the survey was
closed. Sensitivity power analysis showed that the final sample was able to detect single regres-
sion effects with effect sizes of $f^2$ = 0.02 ($t$ = 1.97) with a power of β = .80. Participants read the
informed consent sheet with instructions and the note that they would be allowed to quit the
survey at any point, then started the survey with following sections in fixed order. Median
completion time was 16.9 minutes. The data and analysis script are provided in S1 File and S1
Data; the original items (in German) are provided in S2 File.

### Discrepancy scale

We presented a novel, situational discrepancy questionnaire, consisting of two main facets:
motivational-affective discrepancy, containing items regarding the frustration of control [31]
(e.g., "Because of the Coronavirus, what happens in my life is currently beyond my control"; 4
items), self-efficacy [32] (e.g., "Even if I invest the necessary effort, I cannot solve the problems
that arise with the coronavirus"; 4 items), and uncertainty [33] (e.g., "The uncertainty sur-
rounding the Coronavirus keeps me from living a fulfilled life", 4 items). These items are con-
ceptually related to the view that the obstruction of goal-attainment induces threat states [8].
The second aspect was a three-item subscale to measure epistemic violations of expectancies

during the coronavirus pandemic (e.g., "The development of the corona pandemic took me by surprise"). A confirmatory factor analysis supported a bifactor solution of a *motivational-affective discrepancy* (autonomy, agency, and uncertainty), and an *epistemic discrepancy* scale, $\chi^2$ (71) = 185.52; RMSEA = .064; SRMR = .053; CFI = .928. The latent variables *Motiv.Discrepancy* and *Epist.Discrepancy* were therefore used for subsequent analyses.

## State affect

Next, we assessed state affect in 34 items judged on a 5-point likert-type scale. Central to our assumptions, we generated three latent variables for motivational-affective states: *BIS-anxiety*, consisting of PANAS-X-fear and BIS-state items (8 items, e.g., *anxious*, *inhibited*, *worried*) [34–36]; *reactance affect*, consisting of reactance and PANAS-X-hostility items (13 items, e.g. *restricted*, *frustrated*, *hostile*) [37, 38]; and *BAS-approach*, consisting of BAS-high activation items (5 items; e.g., *determined*, *energetic*) [36, 39]. The CFA for the three latent variables showed acceptable fit ($\chi^2$ (270) = 674.59; RMSEA = .062; SRMR = .067; CFI = .924). For additional exploratory purposes, we also measured deactivated BAS-affect (e.g., *relaxation*) and sadness, which were not included in the analysis.

## Action judgments

In the next section of the questionnaire, participants were presented thirteen concrete action strategies. In three Likert-type items (scaled 1–7), they were instructed to rate to what extent a) they thought the strategy was sensible, b) they thought the strategy could help them feel better, and c) they were planning to engage in this behavior in the following days. The original German items can be found in the S2 File. The three rating items showed high internal consistency ($\alpha$ = .92) and were combined to a single indicator of attractivity for subsequent analyses.

To reduce dimensionality for structural equation analyses, we reduced the actions to three components of interest: Pursuing personal projects (abstract personal *self-enhancement*: personal projects, doing sports, and being creative); social media use (concrete social: using social media, keeping and re-establishing social contacts through digital media, and repeatedly looking for information online), and security efforts (concrete personal: washing hands, stockpiling, and disregarding curfews [reverse-coded]). We entered the assumed components as latent factors in a confirmatory factor analysis ($\chi^2$ (18) = 31.09; RMSEA = .043; SRMR = .040; CFI = .979) and used the latent variables *personal projects*, *social media use*, and *security efforts* for the structural equation analysis. The correlations between the measures can be seen in Table 1.

## System justification

We adapted the system justification questionnaire [15] to investigate the perception of the socio-political system in the COVID-19 situation as fair (e.g., *in this crisis, the political system functions as it should*). The eight-item scale showed good fit in a single-factor CFA, $\chi^2$ (12) = 30.46; RMSEA = .062; SRMR = .036; CFI = .972.

## Loneliness

As an additional predictor, we assessed loneliness [40] with a 15-item scale. The scale contains the subscales social (*I don't have any friends who share my views, but I wish I did*), family (*I feel alone when I am with my family*), and romantic (*I have a romantic partner to whose happiness I contribute*, reverse-coded) loneliness, but the items loaded on a single-factor indicator, as indicated by the CFA ($\chi^2$ (60) = 94.73; RMSEA = .038; SRMR = .038; CFI = .992).

**Table 1. Means, standard deviations, and correlations with confidence intervals of tested variables.**

| Variable | M | SD | 1 | 2 | 3 | 4 | 5 | 6 | 7 | 8 | 9 | 10 | 11 | 12 | 13 | 14 | 15 |
|---|---|---|---|---|---|---|---|---|---|---|---|---|---|---|---|---|---|
| 1. Motiv. Discr | 2.93 | 0.58 | | | | | | | | | | | | | | | |
| 2. Exp. Discr. | 3.22 | 0.93 | .10 [-.00, .19] | | | | | | | | | | | | | | |
| 3. BIS | 2.22 | 0.81 | .89** [.87, .91] | .12* [.02, .21] | | | | | | | | | | | | | |
| 4. BAS | 2.98 | 0.78 | -.80** [-.83, -.76] | -.15** [-.24, -.05] | -.67** [-.72, -.61] | | | | | | | | | | | | |
| 5. Reactance | 1.87 | 0.53 | .89** [.86, .91] | .11* [.01, .20] | .83** [.80, .86] | -.68** [-.73, -.63] | | | | | | | | | | | |
| 6. Loneliness | 5.56 | 1.01 | .40** [.32, .48] | .05 [-.04, .15] | .29** [.20, .38] | -.43** [-.51, -.35] | .35** [.26, .44] | | | | | | | | | | |
| 7. Own Projects | 3.74 | 0.73 | -.26** [-.35, -.17] | .08 [-.02, .18] | -.17** [-.26, -.07] | .26** [.16, .35] | -.07 [-.17, .03] | -.22** [-.31, -.13] | | | | | | | | | |
| 8. Media Use | 2.90 | 0.63 | .38** [.29, .46] | .12* [.02, .22] | .49** [.41, .56] | -.06 [-.16, .04] | .35** [.26, .43] | -.09 [-.19, .01] | -.24** [-.33, -.14] | | | | | | | | |
| 9. Security Behaviors | 3.40 | 0.47 | .11* [.01, .21] | -.08 [-.17, .02] | .31** [.22, .40] | .06 [-.04, .16] | .01 [-.09, .11] | -.38** [-.47, -.30] | .18** [.08, .27] | .62** [.55, .67] | | | | | | | |
| 10. System Justification | 4.72 | 0.95 | -.25** [-.34, -.16] | .18** [.09, .28] | -.22** [-.31, -.12] | .23** [.13, .32] | -.31** [-.39, -.21] | -.41** [-.49, -.33] | .15** [.05, .24] | .25** [.15, .34] | .04 [-.06, .14] | | | | | | |
| 11. Age | 34.40 | 14.82 | -.08 [-.18, .02] | -.03 [-.13, .07] | -.09 [-.19, .00] | .13* [.03, .22] | -.19** [-.28, -.09] | -.17** [-.26, -.07] | -.22** [-.31, -.13] | -.07 [-.17, .03] | .04 [-.06, .14] | -.04 [-.13, .06] | | | | | |
| 12. Gender | 0.68 | 0.47 | .09 [-.01, .19] | .09 [-.01, .19] | .13** [.04, .23] | -.09 [-.19, .01] | .02 [-.08, .12] | -.03 [-.13, .07] | -.03 [-.13, .07] | .24** [.15, .33] | .26** [.16, .35] | -.00 [-.10, .10] | -.11* [-.21, -.01] | | | | |
| 13. Country | 0.32 | 0.47 | -.07 [-.17, .03] | .13** [.03, .23] | -.07 [-.17, .03] | .05 [-.05, .15] | -.03 [-.13, .07] | .09 [-.01, .19] | .11* [.01, .21] | .00 [-.09, .10] | -.07 [-.17, .02] | .07 [-.03, .17] | -.27** [-.36, -.18] | .10* [.00, .20] | | | |
| 14. Risk Group | 0.13 | 0.34 | .14** [.05, .24] | -.14** [-.23, -.04] | .11* [.01, .20] | -.07 [-.17, .03] | .03 [-.07, .12] | .05 [-.05, .15] | -.21** [-.31, -.12] | .03 [-.06, .13] | .07 [-.03, .17] | -.12* [-.22, -.02] | .33** [.24, .42] | .04 [-.06, .14] | -.13** [-.23, -.04] | | |
| 15. Precondition | 0.12 | 0.33 | .06 [-.04, .16] | -.08 [-.18, .02] | .04 [-.06, .14] | -.02 [-.12, .08] | .01 [-.09, .11] | -.02 [-.12, .07] | -.09 [-.19, .01] | .05 [-.05, .15] | .08 [-.02, .17] | -.06 [-.16, .04] | .15** [.05, .24] | -.04 [-.14, .06] | -.07 [-.17, .03] | .57** [.50, .64] | |
| 16. Education Degree | 0.44 | 0.50 | -.12* [-.21, -.02] | .03 [-.07, .13] | -.05 [-.15, .04] | .13** [.03, .23] | -.12* [-.22, -.02] | -.1** [-.20, -.01] | .04 [-.06, .14] | .04 [-.06, .14] | .09 [-.01, .16] | .07 [-.03, .17] | .25** [.16, .34] | -.10* [-.20, -.01] | -.06 [-.16, .04] | -.10 [-.20, .00] | -.00 [-.10, .10] |

*Note*. M and SD represent mean and standard deviation, respectively. They were calculated by averaging the (correctly coded) items. Intercorrelations were calculated using the latent variables from the structural equation model. Values in square brackets indicate the 95% confidence interval for each pearson correlation. The confidence interval is a plausible range of population correlations that could have caused the sample correlation (Cumming, 2014).

* indicates $p < .05$.

** indicates $p < .01$. Risk Group and Precondition are binary (0 = false, 1 = true). Gender is coded female = 1, male = 0. Country = country of residence (Austria = 1, Germany = 0). Education Degree is coded (1 = university degree; 0 = no degree).

Lastly, participants provided demographic data, namely age, gender, nationality, country of residence, education level, and number of people in their household. We also asked whether participants considered themselves as part of a risk group, and whether they had pre-existing respiratory conditions (e.g., asthma). Participants were also presented a ten-item knowledge test about the coronavirus ($M$ = 78% correct) and were asked to guess how many of 100,000 people in their region were currently infected with COVID-19. Answers ranged from 0 to 60,000, with a median of 80. Since these were exploratory measures and not relevant to our central hypotheses, they were not included in the analyses.

## Results

### Analysis strategy

To explain the paths of motivational and epistemic discrepancy, emotional state, and the defensive DVs, we calculated a structural equation path model using lavaan version 0.6–5 [41] in R version 3.6.0. Structural equation model parameters were estimated via maximum likelihood method. Indirect effects of the mediating affects were calculated with 1,000 bootstrap samples (95% confidence intervals). The assumptions and hypotheses were not preregistered due to the short-term conception and deployment of the study. However, we report all measures and procedures included in the survey.

We expected both motivational-affective discrepancy and epistemic discrepancy perception to predict affective states of BIS-related anxiety, BAS-related approach motivation, and reactance affect. In a second step, we expected BIS to be related to higher preference for security-related behaviors and system justification, BAS to relate to preference for pursuing personal projects, and reactance affect to relate to lower system justification and lower ratings of security-related strategies.

To account for potential confounds and control for systematic differences between demographic measures, we used the variables age, gender, country of residence, risk group status, and education (degree vs. no degree) as covariates of no interest. To contain the complexity of the model, we specified the respective demographic variables as covariates of variables that were correlated (see Table 1). The model without these covariates is provided in the supplementary materials S3 File. The directions and effect sizes were largely equal for both approaches.

The model fit was acceptable (CFI = .869; RMSEA = .041; SRMR = .067). The model and unstandardized regression coefficients are depicted in Fig 1. Regression weights, mediation paths, and confidence intervals are reported in Tables 2 (regression parameters) and 3 (indirect effects).

The results revealed a significant effect of motivation discrepancy perception on BIS-affect ($b$ = 1.38, $SE$ = 0.18, $p$ < .001, 95% CI [1.03, 1.74]) and reactance ($b$ = 1.21, $SE$ = 0.18, $p$ < .001, 95% CI [0.86, 1.55]), and a significant negative effect on BAS-affect ($b$ = -1.24, $SE$ = 0.17, $p$ < .001, 95% CI [-1.58, -0.90]). Epistemic discrepancy was not significantly related to either affect facet (all $|z|$ <. 1.38; $p$ > .17). Next, we tested the assumptions that the ratings of the defensive strategies would be associated with specific affective states.

### Security-related actions

The preference for security-related strategies (washing hands, stockpiling, less disregard for curfews) was significantly associated with BIS-anxiety ($b$ = 0.37, $SE$ = 0.11, $p$ = .001, 95% CI [0.16, 0.58]), and significantly related to lower reactance scores ($b$ = -0.26, $SE$ = 0.11, $p$ = .014, 95% CI [-0.47, -0.05]). Indirect effects suggested a mediating effect of BIS-anxiety on the relation of motivational discrepancies and security-related behaviors (indirect effect: $b$ = 0.51,

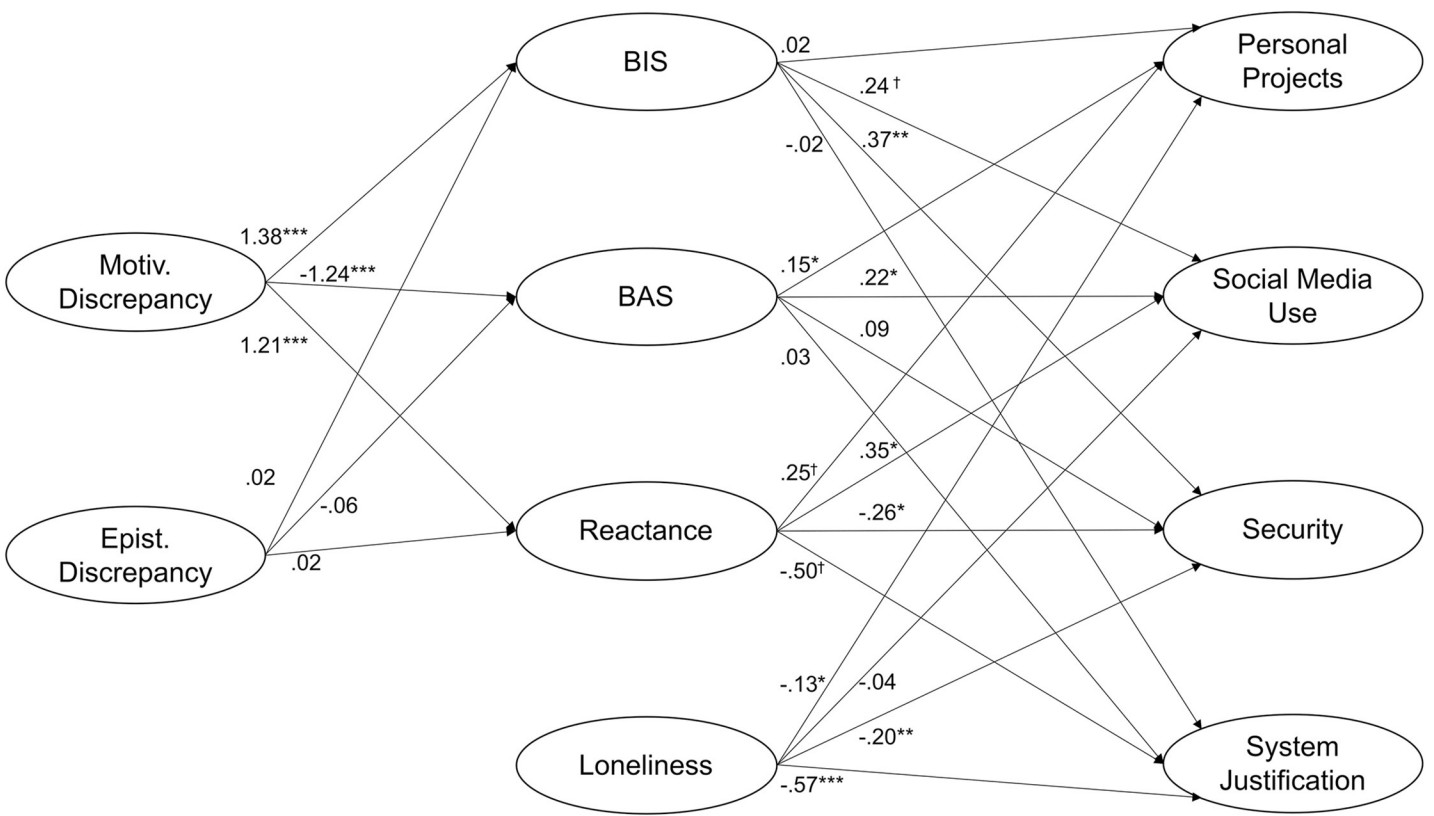

**Fig 1. Structural equation model.** $\chi^2$/df = 1.67, CFI = .869; RMSEA = .041; SRMR = .067. Note: The model shows the unstandardized regression coefficients. For regression and mediation paths, see Tables 2 and 3. BIS: behavioral inhibition system-related anxiety; BAS: behavioral activation system-related approach-affect. * $p <$ .05; ** $p <$ .01; * $p <$ .001.

$SE$ = 0.16, $p$ = .001, 95% CI [0.20, 0.83]), and a significant mediation effect of reactance on decreased security-related behavior ratings ($b$ = -0.32, $SE$ = 0.13, $p$ = .017, 95% CI [-0.58, -0.06]). Loneliness was related to decreased preference for security behavior ($b$ = -0.20, $SE$ = 0.06, $p$ = .002, 95% CI [-0.32, -0.07]).

## Personal projects

In line with our expectations, BAS-related approach was associated with ratings of personal abstract defenses ($b$ = 0.15, $SE$ = 0.07, $p$ = .044, 95% CI [0.00, 0.30]). BAS also mediated the relation between motivational discrepancies and personal projects ($b$ = -0.19, $SE$ = 0.09, $p$ = .050, 95% CI [-0.37, 0.00]). In addition, reactance was marginally related to the ratings of personal projects ($b$ = 0.25, $SE$ = 0.13, $p$ = .054, 95% CI [0.00, 0.49]). The analysis of indirect effects indicated that reactance marginally mediated the interest in personal projects ($b$ = 0.30, $SE$ = 0.16, $p$ = .061, 95% CI [-0.01, 0.61]). Loneliness was associated with lower interest in personal concrete strategies ($b$ = -0.13, $SE$ = 0.06, $p$ = .030, 95% CI [-0.25, -0.01]).

## Social media use

Ratings for social media use were significantly associated with BAS affect ($b$ = 0.22, $SE$ = 0.10, $p$ = .022, 95% CI [0.03, 0.41]). Indirect effects showed that the effect of BAS-approach mediating the association of motivational discrepancy to social media use ratings was also significant ($b$ = -0.27, $SE$ = 0.12, $p$ = .027, 95% CI [-0.52, -0.03]). Social media use was also related to

**Table 2. Regression weights and 95% confidence intervals, based on 1,000 bootstrap samples.**

| Regression | b | SE | z-value | p-value | 95% CI |
|---|---|---|---|---|---|
| Motivational Discrepancy--> BIS | 1.38 | 0.18 | 7.58 | 0.000 | [1.03, 1.74] |
| Motivational Discrepancy--> BAS | -1.24 | 0.17 | -7.08 | 0.000 | [-1.58, -0.90] |
| Motivational Discrepancy--> Reactance | 1.21 | 0.18 | 6.89 | 0.000 | [0.86, 1.55] |
| Epistemic Discrepancy--> BIS | 0.02 | 0.04 | 0.67 | 0.503 | [-0.05, 0.09] |
| Epistemic Discrepancy--> BAS | -0.06 | 0.04 | -1.38 | 0.167 | [-0.15, 0.03] |
| Epistemic Discrepancy--> Reactance | 0.02 | 0.03 | 0.55 | 0.582 | [-0.05, 0.08] |
| BIS--> Personal Projects | 0.02 | 0.10 | 0.18 | 0.856 | [-0.17, 0.21] |
| BAS--> Personal Projects | 0.15 | 0.07 | 2.01 | 0.044 | [0.00, 0.30] |
| Reactance--> Personal Projects | 0.25 | 0.13 | 1.92 | 0.054 | [0.00, 0.49] |
| Loneliness--> Personal Projects | -0.13 | 0.06 | -2.17 | 0.030 | [-0.25, -0.01] |
| BIS--> Social Media Use | 0.24 | 0.13 | 1.77 | 0.076 | [-0.02,0.50] |
| BAS--> Social Media Use | 0.22 | 0.10 | 2.29 | 0.022 | [0.03, 0.41] |
| Reactance--> Social Media Use | 0.35 | 0.15 | 2.31 | 0.021 | [0.05, 0.65] |
| Loneliness--> Social Media Use | -0.04 | 0.08 | -0.45 | 0.651 | [-0.19, 0.12] |
| BIS--> Security | 0.37 | 0.11 | 3.46 | 0.001 | [0.16, 0.58] |
| BAS--> Security | 0.09 | 0.06 | 1.47 | 0.140 | [-0.03, 0.21] |
| Reactance--> Security | -0.26 | 0.11 | -2.47 | 0.014 | [-0.47, -0.05] |
| Loneliness--> Security | -0.20 | 0.06 | -3.10 | 0.002 | [-0.32, -0.07] |
| BIS--> System Justification | -0.02 | 0.23 | -0.07 | 0.945 | [-0.46, 0.43] |
| BAS--> System Justification | 0.03 | 0.17 | 0.17 | 0.869 | [-0.30, 0.36] |
| Reactance--> System Justification | -0.50 | 0.27 | -1.84 | 0.065 | [-1.03, 0.03] |
| Loneliness--> System Justification | -0.57 | 0.15 | -3.71 | 0.000 | [-0.87, -0.27] |
| Motivational Discrepancy--> Personal Projects | -0.38 | 0.26 | -1.48 | 0.138 | [-0.89, 0.12] |
| Motivational Discrepancy--> Social Media Use | -0.32 | 0.32 | -0.99 | 0.323 | [-0.94, 0.31] |
| Motivational Discrepancy--> Security | 0.07 | 0.20 | 0.36 | 0.722 | [-0.32, 0.47] |
| Motivational Discrepancy--> System Justification | 0.24 | 0.56 | 0.43 | 0.665 | [-0.86, 1.34] |
| Epistemic Discrepancy--> Personal Projects | 0.05 | 0.04 | 1.34 | 0.181 | [-0.02, 0.13] |
| Epistemic Discrepancy--> Social Media Use | 0.03 | 0.05 | 0.52 | 0.601 | [-0.08, 0.13] |
| Epistemic Discrepancy--> Security | -0.05 | 0.03 | -1.45 | 0.148 | [-0.12, 0.02] |
| Epistemic Discrepancy--> System Justification | 0.27 | 0.10 | 2.81 | 0.005 | [0.08, 0.46] |
| Gender--> BIS | 0.16 | 0.05 | 3.17 | 0.002 | [0.06, 0.26] |
| Gender--> Social Media Use | 0.45 | 0.08 | 5.56 | 0.000 | [0.29, 0.61] |
| Gender--> Security | 0.19 | 0.06 | 3.08 | 0.002 | [0.07, 0.31] |
| Risk Group--> BIS | 0.02 | 0.07 | 0.22 | 0.829 | [-0.12, 0.15] |
| Risk Group--> Personal Projects | -0.17 | 0.09 | -1.96 | 0.049 | [-0.34, 0.00] |
| Risk Group--> System Justification | -0.35 | 0.20 | -1.72 | 0.086 | [-0.75, 0.05] |
| Education Level--> BAS | 0.08 | 0.07 | 1.20 | 0.232 | [-0.05, 0.21] |
| Education Level--> Reactance | -0.02 | 0.05 | -0.53 | 0.597 | [-0.12, 0.07] |
| Age--> BAS | 0.01 | 0.00 | 2.37 | 0.018 | [0.00, 0.01] |
| Age--> react | -0.01 | 0.00 | -4.49 | 0.000 | [-0.01, 0.00] |
| Age--> Personal Projects | -0.01 | 0.00 | -4.08 | 0.000 | [-0.01, -0.01] |
| Country_AT--> Personal Projects | 0.06 | 0.06 | 1.04 | 0.297 | [-0.06, 0.18] |

reactance ($b$ = 0.35, $SE$ = 0.15, $p$ = .021, 95% CI [0.05, 0.65]), and reactance mediatfed the effect of motivational discrepancy on media usage ($b$ = 0.42, $SE$ = 0.19, $p$ = .025, 95% CI [0.05, 0.80]). Thus, both approach-related BAS states and reactance were related to concrete personal behavior intentions.

**Table 3. Indirect effect regression weights and 95% confidence intervals (1,000 bootstrap samples).**

| Effect | b | SE | z-value | p-value | 95% CI |
|---|---|---|---|---|---|
| Motivational Discrepancy--> BIS--> Personal Projects | 0.02 | 0.13 | 0.18 | 0.856 | [-0.24, 0.29] |
| Motivational Discrepancy--> BIS--> Social Media Use | 0.33 | 0.19 | 1.73 | 0.083 | [-0.04, 0.70] |
| Motivational Discrepancy--> BIS--> Security | 0.51 | 0.16 | 3.21 | 0.001 | [0.20, 0.83] |
| Motivational Discrepancy--> BIS--> System Justification | -0.02 | 0.32 | -0.07 | 0.945 | [-0.64, 0.60] |
| Motivational Discrepancy--> BAS--> Personal Projects | -0.19 | 0.09 | -1.96 | 0.050 | [-0.37, 0.00] |
| Motivational Discrepancy--> BAS--> Social Media Use | -0.27 | 0.12 | -2.21 | 0.027 | [-0.52, -0.03] |
| Motivational Discrepancy--> BAS--> Security | -0.11 | 0.08 | -1.45 | 0.147 | [-0.27, 0.04] |
| Motivational Discrepancy--> BAS--> System Justification | -0.03 | 0.21 | -0.17 | 0.869 | [-0.45, 0.38] |
| Motivational Discrepancy--> Reactance--> Personal Projects | 0.30 | 0.16 | 1.87 | 0.061 | [-0.01, 0.61] |
| Motivational Discrepancy--> Reactance--> Social Media Use | 0.42 | 0.19 | 2.24 | 0.025 | [0.05, 0.80] |
| Motivational Discrepancy--> Reactance--> Security | -0.32 | 0.13 | -2.39 | 0.017 | [-0.58, -0.06] |
| Motivational Discrepancy--> Reactance--> System Justification | -0.60 | 0.33 | -1.81 | 0.070 | [-1.26, 0.05] |
| Epistemic Discrepancy--> BIS--> Personal Projects | 0.00 | 0.00 | 0.18 | 0.860 | [0.00, 0.01] |
| Epistemic Discrepancy--> BIS--> Social Media Use | 0.01 | 0.01 | 0.63 | 0.529 | [-0.01, 0.02] |
| Epistemic Discrepancy--> BIS--> Security | 0.01 | 0.01 | 0.66 | 0.512 | [-0.02, 0.03] |
| Epistemic Discrepancy--> BIS--> System Justification | 0.00 | 0.01 | -0.07 | 0.946 | [-0.01, 0.01] |
| Epistemic Discrepancy--> BAS--> Personal Projects | -0.01 | 0.01 | -1.13 | 0.258 | [-0.03, 0.01] |
| Epistemic Discrepancy--> BAS--> Social Media Use | -0.01 | 0.01 | -1.18 | 0.237 | [-0.04, 0.01] |
| Epistemic Discrepancy--> BAS--> Security | -0.01 | 0.01 | -1.02 | 0.307 | [-0.02, 0.01] |
| Epistemic Discrepancy--> BAS--> System Justification | 0.00 | 0.01 | -0.16 | 0.870 | [-0.02, 0.02] |
| Epistemic Discrepancy--> Reactance--> Personal Projects | 0.00 | 0.01 | 0.53 | 0.594 | [-0.01, 0.02] |
| Epistemic Discrepancy--> Reactance--> Social Media Use | 0.01 | 0.01 | 0.54 | 0.591 | [-0.02, 0.03] |
| Epistemic Discrepancy--> Reactance--> Security | 0.00 | 0.01 | -0.54 | 0.589 | [-0.02, 0.01] |
| Epistemic Discrepancy--> Reactance--> System Justification | -0.01 | 0.02 | -0.52 | 0.600 | [-0.04, 0.03] |

## System justification

System justification was marginally negatively related to reactance ($b$ = -0.50, $SE$ = 0.27, $p$ = .065, 95% CI [-1.03, 0.03]), and significantly negatively related to loneliness ($b$ = -0.57, $SE$ = 0.15, $p$ < .001, 95% CI [-0.87, -0.27]). Thus, reactance and loneliness were related to lower justification of the socio-political system. However, the indirect effect of reactance as mediator between motivational-affective discrepancy and system justification was not significant ($b$ = -0.60, $SE$ = 0.33, $p$ = .070, 95% CI [-1,26, 0.05]). Notably, epistemic discrepancy showed a significant direct effect on system justification, $b$ = 0.27, $SE$ = 0.10, $p$ = .005, 95% CI [0.08, 0.46]. This effect was not mediated through either affect measure.

## Discussion and outlook

In this empirical study, we aimed to provide insights into what makes the COVID-19 situation threatening, and how individuals' motivational-affective states relate to preferences for defensive strategies. Since we were able to collect data within the first days of COVID-19 restrictions in March 2020, this study provides valuable insight into the early perception of the pandemic and the measures taken among individuals in Germany and Austria. Our results suggest that individual preferences for defensive strategies are influenced by the kind of motivational-affective state elicited by motivational-affective discrepancies, rather than epistemic discrepancy. This is notable, as it suggests that this crisis has enormous motivational forces on security- and growth-focused behaviors. The divergence between motivational and epistemic factors might

have to do with the concreteness of both the threat and the possible defensive strategies–compared to existential concerns that cannot be definitively resolved.

Individual perceptions of the situation and motivational-affective states were related to different behavioral intentions. Personal concrete defenses, measured by ratings of security-related actions, were related to BIS-anxiety and negatively related to reactance and loneliness. The mediating effect of BIS-anxiety was in line with our predictions since these actions serve to reduce concrete infection risk and provide behavioral prescriptions to cling to, thus mitigating anxiety.

Looking at abstract personal strategies, we found the expected relation between approach-motivated BAS state and personal project ratings. Individuals in an approach-related state were more inclined to engage in self-enhancing activities, presumably to reactivate approach behaviors in uncertain situations [8]. For social defenses, there was a statistically significant effect of approach-related BAS affect mediating the connection between motivational discrepancy and social media use. This suggests that approach-oriented individuals were more interested in (re-)establishing social contacts and information gathering through digital media.

On the abstract side of social defenses, epistemic discrepancy in the COVID-19 situation was directly related to increased system justification. This effect was not mediated by either affect component. In our view, this suggests that the affective responses to COVID-19 were mostly related to thwarted motivational-affective needs, not so much to epistemic motives. Epistemic discrepancies seemed to not follow a motivational-affective route but were directly related to abstract social defenses. This is in line with research suggesting that individuals cope with lacking personal control by supporting broader external systems [42, 43]. Since the discrepancies were assessed in a quite concrete context, this might have made the situational aspects of the pandemic more salient than the underlying existential concerns. Future research should investigate whether these paths replicate for epistemic concerns in concrete situations, and whether they translate to scenarios with more abstract existential concerns, for instance in epistemically uncertain situations where individuals need to create meaning [9].

In addition to BIS and BAS states, our study included an assessment of reactance. This is due to the unique situation of this crisis, where "classical" psychological threat concerns and situational pressures create a kind of *super-threat* [7]. In Germany and Austria, rather drastic actions were taken in a short time, which has certainly contained the spread of the coronavirus, but was accompanied by significant restrictions of individual freedom. In line with our assumptions, mediation paths showed that reactance was strongly related to the intentions of pursuing various defensive strategies. Personal abstract defenses were marginally related to a state of reactance. In our view, individuals try to restore personal freedom by pursuing idiosyncratic goals. Inversely, reactance was related to a devaluation of security behaviors which further restrict personal freedom. Reactance was also related to lower system justification, suggesting that if the experience of discrepancy elicited reactance, individuals were more unhappy with the system (and in turn, the politicians imposing the restrictions). Overall, reactance appeared to be a general catalyst for behaviors in response to COVID-19 uncertainty.

In contrast to the motivational fuel provided by reactance, loneliness was generally related to inaction. We found that loneliness was associated with lower interest in pursuing personal goals, lower intentions of following security-related behaviors, and decreased system justification. In sum, being lonely appeared to be big risk factor for passivity and societally maladaptive attitudes. Current research provides further evidence for the detrimental effects of loneliness on mental health [25, 27, 44].

On a personal level, motivational-affective states impact personal behavior directly in the form of media usage and personal projects started. On a societal level, the mediation analyses

on system justification and security effort try to explain why some people show increased appreciation of governments, while others increasingly distrust the decision-makers. For governments, the actions taken to reduce large-scale infection risks and overburdening of the healthcare system comes with a cost, namely the frustration and dissatisfaction of citizens who feel wrongfully restricted. This bears the necessity for political and social measures to be adopted to psychological needs. Future efforts should be made to investigate whether these effects are stable in time or underlie distortions and changes with the progression of curfews and restrictions.

Of course, the survey had its limitations that should be respected when making inferences. The study was a correlative, cross-sectional design without experimental manipulations, causal inferences should thus be made with caution. We decided against an experimental manipulation of corona as a threat or manipulating the salience of defensive strategies for two main reasons: first, we aimed to gauge participants' perceptions of the situation and unmanipulated affective states to adequately represent the general population. Second, even with a neutral control group the COVID situation would have been salient to participants. In a next step, experimental manipulation can be employed to investigate to what extent the salience of a pandemic (vs. a non-threatening control condition) as psychological threat influences the emotional states and preferences for behavioral strategies and attitudes.

To consolidate our findings reported in this manuscript, follow-up research should employ longitudinal designs to further understand the within-subject course of the threat-and-defense process. For instance, an experience-sampling design might shed light on the day-by-day experience of coronavirus threat and its influence on motivational-affective states and subsequent behavioral intentions and attitudes. Further insights into the relation of individual levels of threat experience, concerns, affective states, and behavior intentions are necessary to understand, predict, and shape defensive behaviors in case of further infection waves or other diseases [45]. Potential predictors may lie in personality traits such as BIS and BAS sensitivity which predispose anxious and approach-related reactions to stressors, respectively [46]. In addition, the association between affective states, loneliness, and action preferences should be researched after the restrictions have been lifted and the pandemic is more under control. This would allow better inferences as to whether the reported preferences for behaviors and system justification are general specifically emerging due to the pandemic.

In conclusion, this work provides compelling insights into the process of threat and defense in a real-life setting. In addition to gauging motivational-affective states such as anxiety or reactance [47], we have provided cross-sectional, mediational insights in how these states relate to behavioral intentions during a state of global uncertainty.

## Supporting information

**S1 File. R analysis markdown script.**
(ZIP)

**S2 File. Items and measures (German).**
(DOCX)

**S3 File. R Analysis of structural equation model without covariates.**
(ZIP)

**S1 Data. Experimental data.**
(CSV)

## Acknowledgments

We thank Pauline Reiß and Sophie Schorr for their help in creating the study and collecting data. We further thank Adrian Lueders for his valuable help during data analysis.

## Author Contributions

**Conceptualization:** Stefan Reiss, Vittoria Franchina, Chiara Jutzi, Robin Willardt, Eva Jonas.

**Data curation:** Stefan Reiss.

**Formal analysis:** Stefan Reiss.

**Funding acquisition:** Eva Jonas.

**Investigation:** Chiara Jutzi.

**Methodology:** Vittoria Franchina, Chiara Jutzi, Robin Willardt.

**Resources:** Eva Jonas.

**Supervision:** Eva Jonas.

**Validation:** Stefan Reiss.

**Visualization:** Stefan Reiss.

**Writing – original draft:** Stefan Reiss, Vittoria Franchina, Chiara Jutzi, Robin Willardt, Eva Jonas.

**Writing – review & editing:** Stefan Reiss, Chiara Jutzi, Eva Jonas.

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
