## [Decision Letter · Decision Letter 0]

21 Jul 2020

PONE-D-20-17404

From Anxiety to Action - Experience of Threat, Emotional States, Reactance and Action Preferences in the Early Days of COVID-19 Self-Isolation in Germany and Austria

PLOS ONE

Dear Dr. Reiss,

Thank you for submitting your manuscript to PLOS ONE. After careful consideration, we feel that it has merit but does not fully meet PLOS ONE’s publication criteria as it currently stands. Therefore, we invite you to submit a revised version of the manuscript that addresses the points raised during the review process.

Please find the reviewer's and mine's comments below.

We look forward to receiving your revised manuscript.

Kind regards,

Valerio Capraro

Academic Editor

PLOS ONE

Journal Requirements:

Additional Editor Comments (if provided):

I have now collected one review from one expert in the field. The reviewer likes the paper but suggest several improvements. Therefore, I would like to invite you to revise the paper following the reviewer's comments. I would like to add just one more comment to improve the literature review. The "perspective article" on what social and behavioural science can do to promote pandemic response, published by Van Bavel et al. in Nature Human Behaviour, could be a useful introductory reference.

Van Bavel, J. J., et al. (2020). Using social and behavioural science to support COVID-19 pandemic response. Nature Human Behaviour.

Reviewers' comments:

Reviewer's Responses to Questions

**Comments to the Author**

1. Is the manuscript technically sound, and do the data support the conclusions?

Reviewer #1: Partly

2. Has the statistical analysis been performed appropriately and rigorously? 

Reviewer #1: Yes

3. Have the authors made all data underlying the findings in their manuscript fully available?

Reviewer #1: Yes

4. Is the manuscript presented in an intelligible fashion and written in standard English?

Reviewer #1: Yes

5. Review Comments to the Author

Reviewer #1: This paper analyzes the cross-sectional associations between self-reported Covid-19 threats, self-reported emotional reactions, reactance and loneliness and self-reported action preferences.

The topic of this paper is relevant and the focused mediational pathways are theoretically defensible and interesting. Given the data at hand the contribution is modest but it might still build an interesting basis for further more comprehensive reseach efforts.

In the following I highlight the main issues that would need to be addressed in my view.

1. Research on psychological reaction in the face of the Covid-19 pandemic is currently exploding and there are hundreds of available studies (many published, many more on preprint servers published soon; several research tracker with lists of all of these studies). Based on a detailed search of the literature, I would like to see a more comprehensive (at least some...) embedding of the current study into the relevant state of (social psychological and personality) research on Covid-19. This would be important both because some of your claims can already be substantiated/compared with empirical results and to show how specifically the present research adds to and moves beyond existing research.

2. The analyses are based on cross-sectional associations between a range of self-report measures. The degree to which they are associated can have a multitude of reasons including very generic individual differences in self-concept that are reflected in various of the measures producing the pattern of associations that can (but does not need to be) described as a mediation model. Given the data at hand, the mediational results do not necessarily tell us anything meaningful about underlying processes. This needs to be made crystal clear.

3. There are of course various options to come at least closer to a causal undestanding beyond experimental approaches such as longitudinal designs and the extensive and theoretically backed up control of a large range of potential confounds. Again it need to be made clear that the present design would need to be complemented by such designs before any strong conclusions can be drawn.

4. It is particularly unclear how much the present findings reflect generic associations that would have been found similarly outside of this (or other crises) - of course with other, less specific measures - or whether they indeed reflect something specific about this or other crises. This needs to be discussed and it might be outlined what sort of designs would help to clarify this issue.

5. With regard to mediating processes, one option that is applied regularly (including research on psychological reactions to Covid-19) are daily diary and/or experience-sampling designs in which you would repeatedly assess momentary general states and Covid-19 specific states as well as relevant situational information. This might be discussed as a more optimal way of testing the present hypotheses.

6. Given that the investigated measures share self-concept variance that is covered in broad personality measures the inclsuion of such measures (beyond loneliness) would have been straightforward. This might also allow you to do at least some of the specificity analyses mentioned above.

7. A much more detailed description of the sample would be needed including information on educational and occupational background.

8. Related to the former point: Many of the concrete Covid-19 related stressors and attitudes are unevenly distributed across educational and occupatuional / social status groups. Similar to personality self-concept, these variables can have an effect on all three groups of variables implied in the mediation model (producing the associations that would in this case be wrongly interpreted as mediation). Also, associations could be different depending on such sociodemographic measures. Therefore, a more detailed investigation of the role of these background variables (as covariates/control variables as well as moderators) would be very important.

9. I very much appreciated that the authors made data and code available. For full transparency, I would encourage to additionally (a) include a sentence making clear that the present hypotheses were not preregistered and (b) make explicit that the additional material includes all procedures and measures that were applied independent of whether they were used for the present paper (and add further information to the supplement if needed).

Signed

Mitja Back

6. PLOS authors have the option to publish the peer review history of their article (what does this mean?). If published, this will include your full peer review and any attached files.

Reviewer #1: **Yes: **Mitja Back

---

## [Author Response · Author response to Decision Letter 0]

9 Nov 2020

Responses to the comments of Reviewer #1

Reviewer #1: This paper analyzes the cross-sectional associations between self-reported Covid-19 threats, self-reported emotional reactions, reactance and loneliness and self-reported action preferences.

The topic of this paper is relevant and the focused mediational pathways are theoretically defensible and interesting. Given the data at hand the contribution is modest but it might still build an interesting basis for further more comprehensive reseach efforts.

- Authors’ response: We thank reviewer 1 for the helpful and constructive feedback. We have addressed the remarks as detailed below; we are confident that these changes have significantly improved the overall quality of the paper.

In the following I highlight the main issues that would need to be addressed in my view.

1. Research on psychological reaction in the face of the Covid-19 pandemic is currently exploding and there are hundreds of available studies (many published, many more on preprint servers published soon; several research tracker with lists of all of these studies). Based on a detailed search of the literature, I would like to see a more comprehensive (at least some…) embedding of the current study into the relevant state of (social psychological and personality) research on Covid-19. This would be important both because some of your claims can already be substantiated/compared with empirical results and to show how specifically the present research adds to and moves beyond existing research.

- Authors’ response: We have updated the introduction with current references to psychological research on Covid-19. To compare our results with meanwhile published research, we have also updated the discussion section of the manuscript.

Please note that due to the use of Mendeley as citation software, the changes in references are not tracked in the marked-up version of the manuscript. 

We have added the following references in the revision of the manuscript: 1, 2, 3, 5, 6, 7, 19, 20, 22, 25, 27, 44, 45, 47

2. The analyses are based on cross-sectional associations between a range of self-report measures. The degree to which they are associated can have a multitude of reasons including very generic individual differences in self-concept that are reflected in various of the measures producing the pattern of associations that can (but does not need to be) described as a mediation model. Given the data at hand, the mediational results do not necessarily tell us anything meaningful about underlying processes. This needs to be made crystal clear.

- Authors’ response: We have addressed this point in the discussion section, and clarified that the results of our study are cross-sectional (p.19, p.20). We do derive the hypotheses on mediational effects from previous literature, but of course cannot prove or disprove orders of causality. That said, in concordance with the theoretical background, the data at least support the suggested order of processes.

3. There are of course various options to come at least closer to a causal undestanding beyond experimental approaches such as longitudinal designs and the extensive and theoretically backed up control of a large range of potential confounds. Again it need to be made clear that the present design would need to be complemented by such designs before any strong conclusions can be drawn.

- Authors’ response: We have updated the discussion section (p.19) to better contextualize our findings. We have also emphasized the importance of conducting further experimental research and controlling for potential confounds.

4. It is particularly unclear how much the present findings reflect generic associations that would have been found similarly outside of this (or other crises) - of course with other, less specific measures - or whether they indeed reflect something specific about this or other crises. This needs to be discussed and it might be outlined what sort of designs would help to clarify this issue.

- Authors’ response: We have amended the discussion to include this discussion point (p. 20). To our knowledge, the association between different motivational-affective states and real-life behavioral preference has not yet been investigated in other psychological threat contexts. Due to this, our point is: “the association between affective states, loneliness, and action preferences should be researched after the restrictions have been lifted and the pandemic is more under control. This would allow better inferences as to whether the reported preferences for behaviors and system justification are general specifically emerging due to the pandemic.”

5. With regard to mediating processes, one option that is applied regularly (including research on psychological reactions to Covid-19) are daily diary and/or experience-sampling designs in which you would repeatedly assess momentary general states and Covid-19 specific states as well as relevant situational information. This might be discussed as a more optimal way of testing the present hypotheses.

- Authors’ response: Reviewer 1 suggests a very viable method to gain more insights into the daily experiences of threat during the pandemic. We have added this point in the discussion section (p.19f.) of the manuscript. 

“To consolidate our findings reported in this manuscript, follow-up research should employ longitudinal designs to further understand the within-subject course of the threat-and-defense process. For instance, an experience-sampling design might shed light on the day-by-day experience of coronavirus threat and its influence on motivational-affective states and subsequent behavioral intentions and attitudes. Further insights into the relation of individual levels of threat experience, concerns, affective states, and behavior intentions are necessary to understand, predict, and shape defensive behaviors in case of further infection waves or other diseases [45].”

6. Given that the investigated measures share self-concept variance that is covered in broad personality measures the inclsuion of such measures (beyond loneliness) would have been straightforward. This might also allow you to do at least some of the specificity analyses mentioned above.

- Authors’ response: We thank reviewer 1 for the suggestion and agree that additional personality measures may contribute to better predict defense preferences during the pandemic. However, the measures employed in the reported manuscript were chosen out of specific interest in the context of the research in our lab group. The focus here was rather on momentary affective states and circumstances (loneliness) than on personality traits or predispositions. We think our manuscripts contributes to the research from this perspective. To account for that point, we have added a sentence on the potential contribution of personality traits on the process (p. 20).

“Potential predictors may lie in personality traits such as trait BIS and BAS sensitivity which predispose anxious and approach-related reactions to stressors, respectively [46].”

7. A much more detailed description of the sample would be needed including information on educational and occupational background.

- Authors’ response: We have amended the sample description to present the sample in more detail. For the suggested additional analyses, we have dichotomized the educational status (university degree vs. no university degree). We have not asked participants to provide more specific details about their occupational status or background, hence we cannot speak to this variable’s influence on the relations.

8. Related to the former point: Many of the concrete Covid-19 related stressors and attitudes are unevenly distributed across educational and occupatuional / social status groups. Similar to personality self-concept, these variables can have an effect on all three groups of variables implied in the mediation model (producing the associations that would in this case be wrongly interpreted as mediation). Also, associations could be different depending on such sociodemographic measures. Therefore, a more detailed investigation of the role of these background variables (as covariates/control variables as well as moderators) would be very important.

- Authors’ response: We thank the reviewer for this suggestion. As a first step, we have added a table including descriptives and intercorrelations of all variables included in the model, and additional demographic / socio-economic descriptors of the sample. 

In a second step, we re-ran the analysis and added the variables age, gender (female vs. male), country of residence (Austria vs. Germany), risk group status, and education (university degress vs. no degree) as covariates wherever there were significant correlations with the SEM variables. The model syntax and results can be found in supplementary file S1 File. 

The old model and results have been moved to supplementary file S3 File for transparency. The results consolidate the findings reported in the initial submission, as virtually all indirect (i.e. mediation) effects remain roughly equal in terms of z-value and statistical significance. 

Most indirect effects show similar but slightly larger relations in the model including covariates (e.g. Motivational Discrepancy -> BIS -> Security efforts, from z = 2.03, p = .042 to z = 3.21, p = .001).

9. I very much appreciated that the authors made data and code available. For full transparency, I would encourage to additionally (a) include a sentence making clear that the present hypotheses were not preregistered and (b) make explicit that the additional material includes all procedures and measures that were applied independent of whether they were used for the present paper (and add further information to the supplement if needed). 

Signed

Mitja Back

- Authors’ response: The reviewer raises an important point regarding the preregistration of our hypotheses and the model used. Due to the hasty nature of early restrictions in the coronavirus pandemic, we have regretfully not preregistered our hypotheses. We have clarified this point in the manuscript (p. 11): “The assumptions and hypotheses were not preregistered due to the short-term conception and deployment of the study. However, we report all measures and procedures included in the survey.”

We do however think that the model has been derived from previous threat-and-defense research. 

Regarding the second point, we have now stated explicitly that the supplementary material includes all measures applied, regardless of inclusion in the manuscript.

6. PLOS authors have the option to publish the peer review history of their article (what does this mean?). If published, this will include your full peer review and any attached files.

Do you want your identity to be public for this peer review? For information about this choice, including consent withdrawal, please see our Privacy Policy.

Reviewer #1: Yes: Mitja Back

---

## [Editor Report · Decision Letter 1]

18 Nov 2020

From Anxiety to Action - Experience of Threat, Emotional States, Reactance and Action Preferences in the Early Days of COVID-19 Self-Isolation in Germany and Austria

PONE-D-20-17404R1

Dear Dr. Reiss,

We’re pleased to inform you that your manuscript has been judged scientifically suitable for publication and will be formally accepted for publication once it meets all outstanding technical requirements.

Kind regards,

Valerio Capraro

Academic Editor

PLOS ONE
---

## [Editor Report · Acceptance letter]

23 Nov 2020

PONE-D-20-17404R1 

From anxiety to action - experience of threat, emotional states, reactance, and action preferences in the early days of COVID-19 self-isolation in Germany and Austria 

Dear Dr. Reiss:

I'm pleased to inform you that your manuscript has been deemed suitable for publication in PLOS ONE. Congratulations! Your manuscript is now with our production department. 

Kind regards, 

on behalf of

Dr. Valerio Capraro 

Academic Editor

PLOS ONE